# Measurements of inclusive and differential cross-sections of $t\bar{t}\gamma$ production in $pp$ collisions at $\sqrt{s} = 13$ TeV with the ATLAS detector

**Carmen Diez Pardos on behalf of the ATLAS Collaboration** [1]

Universität Siegen, Siegen, Germany

⋆ carmen.diez.pardos@cern.ch

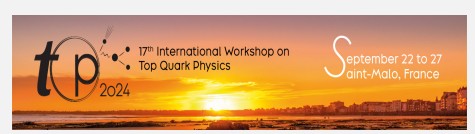

*The 17th International Workshop on
Top Quark Physics (TOP2024)
Saint-Malo, France, 22-27 September 2024*

## Abstract

**Cross-section measurements of the associated production of a top quark pair and a photon ($t\bar{t}\gamma$) are performed with an integrated luminosity of $140\,\text{fb}^{-1}$ of proton–proton collisions at a centre-of-mass energy of 13 TeV collected by the ATLAS detector at the LHC. The measurement focuses on $t\bar{t}\gamma$ topologies where the photon is radiated from an initial-state parton or one of the top quarks. The differential cross-sections are measured for variables characterising the photon, lepton and jet kinematic properties. The distribution of the photon transverse momentum is used to constrain effective field theory operators related to the electroweak dipole moments of the top quark.**

## 1 Introduction

Measurements of top quark pair production ($t\bar{t}$) in association with a photon ($t\bar{t}\gamma$) provide crucial information on the $t - \gamma$ electroweak coupling. Additionally, the process is sensitive to top quark anomalous dipole moments, which can be interpreted in the context of Effective Field Theories (EFT). The photon radiation in $t\bar{t}$ events may originate from the initial-state quarks, the top quarks, or the charged particles from the top quark decay. The publication presented here [1] focuses on the $t\bar{t}\gamma$ processes where the photon is radiated at the production stage ($t\bar{t}\gamma$ production in the following), which is the most sensitive to the $t - \gamma$ coupling. The cross-sections at stable particle level in a fiducial phase space are measured using the data set collected during Run 2 with the ATLAS detector at the LHC [2] corresponding to an integrated luminosity of 140 fb$^{-1}$. Both the single-lepton and dilepton $t\bar{t}$ decay channels are considered.

## 2 Analysis strategy

Events in the single-lepton channel are selected if they contain exactly one photon and one isolated lepton, electron or muon. Events must contain at least four reconstructed jets, and at least one of them is required to be $b$-tagged. In the dilepton channel, events are selected if they contain exactly two oppositely-charged, isolated leptons ($ee$, $e\mu$, $\mu\mu$) and at least two

---

jets out of which at least one is $b$-tagged. Additional kinematic requirements are imposed in both channels to further suppress background processes.

The background processes are classified into several categories depending on whether the photon is prompt or a mis-reconstructed object that mimics the photon. Events with prompt photons can arise from $t\bar{t}\gamma$ with photons from decay and other sources ($W/Z$, single-top quark, diboson and $t\bar{t}$ in association with a $Z$ or $W$ boson) and are estimated using Monte Carlo simulations. Events with misidentified, fake photons are estimated using data-driven methods. The expected contribution from events with mis-reconstructed electrons is estimated using control regions enriched in $Z \rightarrow e\gamma$ and $Z \rightarrow ee$ events. Events with mis-reconstructed jets or photons with hadron origin are estimated using the so-called ABCD method.

Neural networks (NNs) are used to enhance the separation of $t\bar{t}\gamma$ production and the background processes. In the single-lepton channel, a four-class NN is used to define a signal region enriched in $t\bar{t}\gamma$ production events and three control regions enriched in $t\bar{t}\gamma$ decay, photon fakes and other $t\bar{t}\gamma$ events, respectively. In the case of the dilepton channel, a binary classifier is used to separate $t\bar{t}\gamma$ production events from all the backgrounds.

## 3  Cross-section measurements

The cross-sections are measured at particle level in a single-lepton (dilepton) fiducial phase space, requiring exactly one isolated photon with high transverse momentum ($p_T$) and 1 (2) leptons (electron or muon), at least 4 (2) jets and at least one $b$-jet. The angular distances $\Delta R = \sqrt{(\Delta\phi)^2 + (\Delta\eta)^2}$ between the photon and the lepton(s) and between any of the jets and the photon and the leptons must be greater than 0.4. The inclusive cross sections are obtained from a simultaneous profile-likelihood fit in the signal and control regions. The combined cross-section results

$$\sigma_{t\bar{t}\gamma \text{ production}} = 319 \pm 15 \, \text{fb} = 319 \pm 4 \, (\text{stat})^{+15}_{-14}(\text{syst}) \, \text{fb}.$$

The uncertainties are dominated by the modelling of the $t\bar{t}\gamma$ processes. The expected cross-section given by the next-to-leading order MadGraph5_aMC@NLO+Pythia 8 simulation is $296^{+29}_{-30}(\text{scale})^{+6}_{-4}(\text{PDF}) \, \text{fb}$, compatible with the measured value within the uncertainties.

The differential cross-sections are measured as functions of photon kinematic variables, angular separations between the photon, leptons and jets in the event. Representative examples of the absolute $t\bar{t}\gamma$ production cross sections measured in the individual channels and in the combined fiducial phase space are shown in Figure 1.

## 4  EFT interpretation

The results are interpreted in the context of EFT. In particular, the sensitivity of the $t\bar{t}\gamma$ production process to the dipole operators $C_{tB}$, $C_{tW}$, coupling to the weak hypercharge and isospin gauge bosons, respectively, is tested. The limits on the Wilson coefficients in the SM effective field theory (SMEFT) framework [3] are obtained from the differential cross-section measurement of the photon $p_T$. Example results of the two-dimensional marginalised posteriors representing the 68% and 95% credible intervals for two combinations of the real and imaginary parts of the coefficients are shown in Figure 2 (top row). The production of $t\bar{t}$ in association with a $Z$ boson ($t\bar{t}Z$) is also modified by the same operators, providing complementary constraining power. Therefore, the EFT interpretation is also performed by using a simultaneous fit to the photon $p_T$ and the $Z$ boson $p_T$ spectra measured in $t\bar{t}Z$ [4]. The limits are additionally obtained in the rotated ($C_{tZ}, C_{t\gamma}$) basis to gauge the relevance of each individual measurement

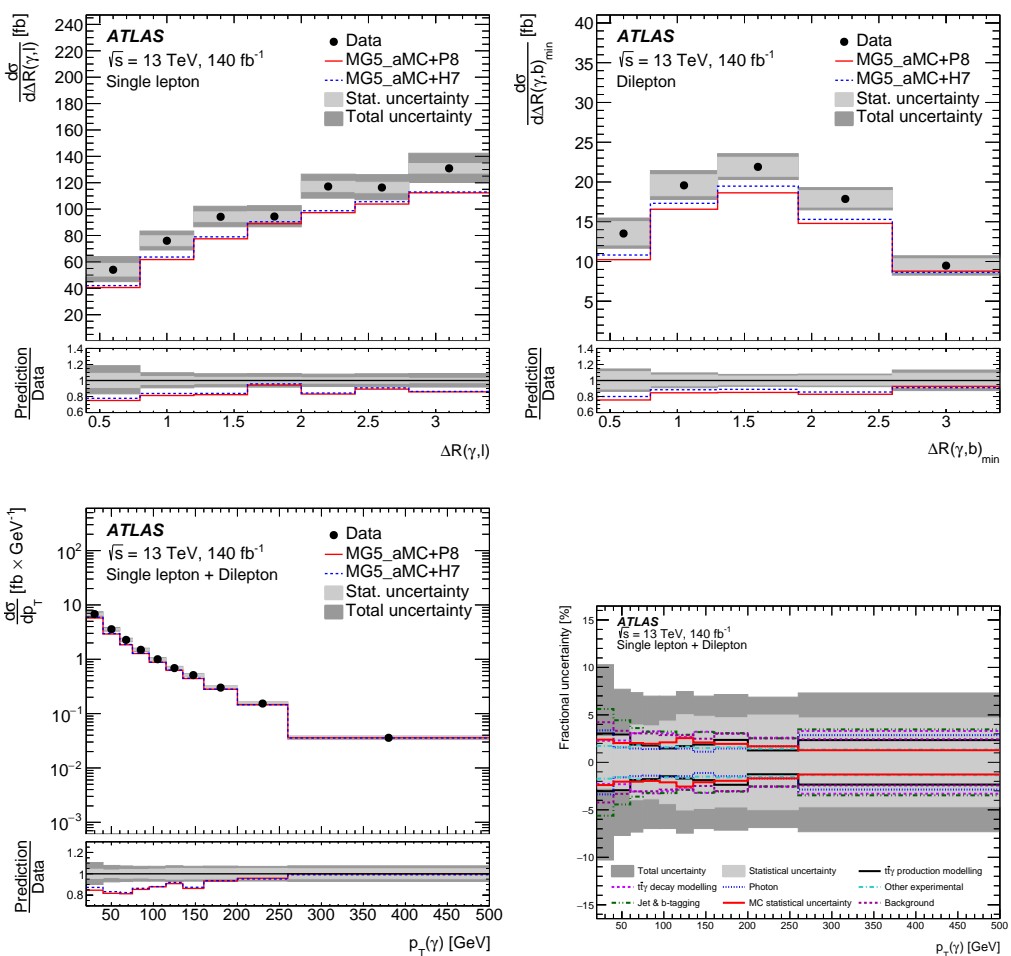

Figure 1: Absolute differential $t\bar{t}\gamma$ production cross sections measured at particle level as a function of $\Delta R(\gamma, \ell)$ in the single-lepton channel (top left), $\Delta R(\gamma, b)_{min}$ in the dilepton channel (top right) and $p_T$ in the combined measurement in the single-lepton and dilepton channels (bottom left) and impact of the systematic uncertainties on the latter variable grouped into different categories [1].

in the combination. Those coefficients describe modifications of the $t\bar{t}\gamma$ interaction vertex and the $t\bar{t}Z$ interaction vertex, respectively. Two illustrative examples of the individual limits and the combination are presented in Figure 2 (bottom). All results are found to be compatible with the SM predictions.

## 5  Conclusion

Inclusive and differential $t\bar{t}\gamma$ cross-section measurements are performed using an integrated luminosity of 140 fb$^{-1}$ of proton–proton collisions at a centre-of-mass energy of 13 TeV, collected by the ATLAS detector at the LHC. The analysis focuses on $t\bar{t}\gamma$ topologies where the photon is radiated from an initial-state parton or one of the top quarks. The results are used to set limits on EFT parameters related to the electroweak dipole moments of the top quark. The EFT interpretation is also performed using a simultaneous fit to the photon $p_T$ and $Z$ boson $p_T$ spectra measured in $t\bar{t}Z$ topologies.

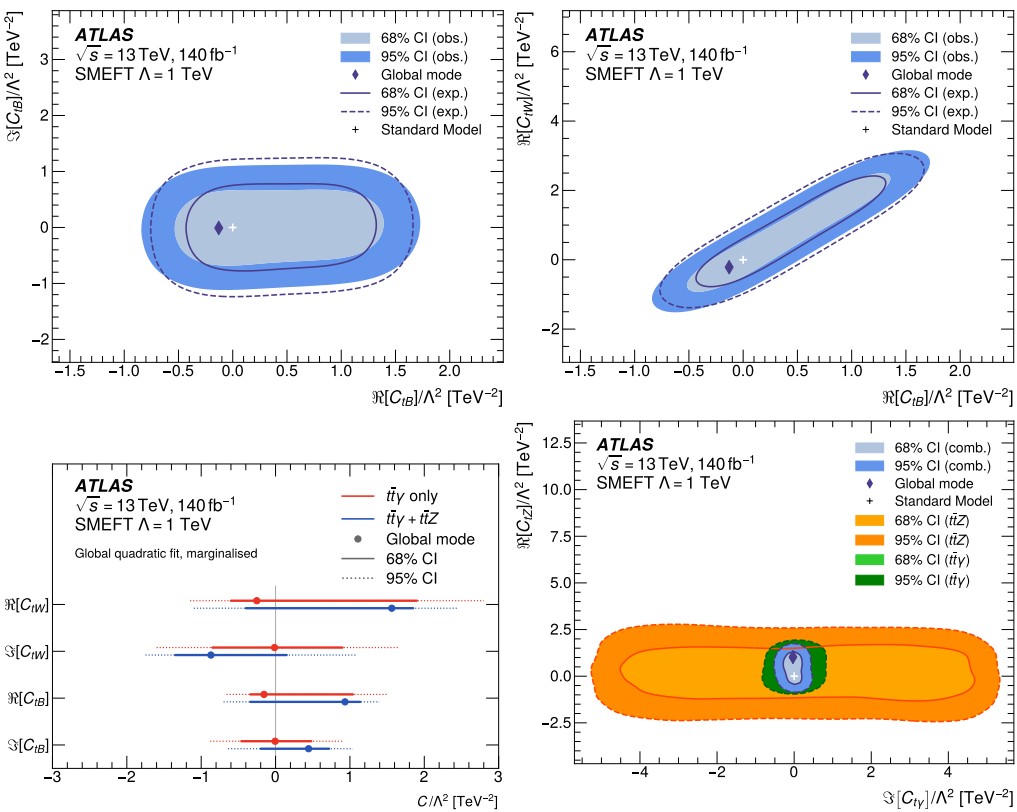

Figure 2: Two-dimensional marginalised posteriors representing the 68% and 95% credible intervals for the pairs of coefficients $\Re[C_{tB}], \Im[C_{tB}]$ (top left) and $\Re[C_{tB}], \Im[C_{tW}]$ (top right). Comparison of the observed 68% and 95% credible intervals for the $C_{tB}$ and $C_{tW}$ operators obtained from the $t\bar{t}\gamma$ measurement and in combination with $t\bar{t}Z$ (bottom left) and the two-dimensional marginalised posteriors for $\Im[C_{t\gamma}], \Re[C_{tZ}]$ from the individual $t\bar{t}\gamma$, $t\bar{t}Z$ measurements and the combined result (bottom right) [1].

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
