# Peer review of "Measurements of inclusive and differential cross-sections of $t\bar{t}γ$ production in $pp$ collisions at $\sqrt{s}=13$ TeV with the ATLAS detector"

_SciPost Physics Proceedings_

## Round 1 · Referee Report · Anonymous (Referee 1) · 2025-1-9

Report

This is a very clear summary of the very interesting and complete analysis of ttγ production in pp collisions at 13 TeV by ATLAS. The impressive set of results presented includes the inclusive and differential cross-section measurements, as well as, an EFT interpretation.

The text is very clear and provides a nice short summary, nevertheless given the shortness of the proceedings few items might be clarified or referenced. For those I add my request and suggestions below, once they are addressed I recommend these proceedings be published.

Requested changes

Analysis strategy: - L1:"Events in the single-lepton channel are selected if they contain exactly one photon and one isolated lepton" -> is the lepton isolated or passing any quality or pT-relevant criteria? Would be good to add a short description such as "one isolated photon"

  • L4: I assume you also request to have at least one photon in this category, at the moment is not explicitly stated.

  • L8: "events" -> "background events".

  • L14: please add a reference for the ABCD method.

  • How do you deal with conversion background? Could a line be added on this?

Cross-section measurements: - L2: "high transverse momentum": can this be quantified?

  • L9: "The uncertainties are dominated by the modelling of the t¯tγ processes.". Does this statement apply only to the inclusive cross-section? From Fig 1 I see Jet and b-tagging uncertainties have a similar size to the modelling ones.

EFT interpretation: - Is this performed using both channels?

Fig 2: There is a mismatch between the top right plot and the caption "I(CtW)" vs "R(CtW)", please correct the caption.

Recommendation

Ask for minor revision

  • validity: -
  • significance: -
  • originality: -
  • clarity: -
  • formatting: -
  • grammar: -

Author:  Carmen Diez Pardos  on 2025-04-24  [id 5413]

(in reply to Report 1 on 2025-01-09)

Dear referee,

thank you for reviewing the draft and the comments. Please find below some detailed replies summarising the changes in the resubmitted draft.

All the best, Carmen Diez Pardos

Analysis strategy: - L1:"Events in the single-lepton channel are selected if they contain exactly one photon and one isolated lepton" -> is the lepton isolated or passing any quality or pT-relevant criteria? Would be good to add a short description such as "one isolated photon" ==> Yes, photons and leptons are isolated and all objects pass quality, pT and eta criteria. For all those details please refer to the original publication, doi:10.1007/JHEP10(2024)191. Added the clarification that is an isolated photon.

  • L4: I assume you also request to have at least one photon in this category, at the moment is not explicitly stated. ==> Yes, rephrased: "Events with exactly one isolated photon are selected. In the single-lepton channel, events must contain exactly one isolated lepton [...]"

  • L8: "events" -> "background events". ==> Modified as suggested

  • L14: please add a reference for the ABCD method. ==> Added, the same as in the original publication

  • How do you deal with conversion background? Could a line be added on this? ==> There is no explicit background category as "conversion background". If the comment refers to fake leptons, this contribution is ~1% (negligible in the dilepton channel), see original publication. For the proceedings, I would rather keep the focus on the main backgrounds.

Cross-section measurements: - L2: "high transverse momentum": can this be quantified? ===> For simplicity and the recommended length of the proceedings for a poster contribution, no detailed pt, eta cuts, etc. are given. Please refer to the publication for the full description of the objects at particle level and the exact requirements.

  • L9: "The uncertainties are dominated by the modelling of the t¯tγ processes.". Does this statement apply only to the inclusive cross-section? From Fig 1 I see Jet and b-tagging uncertainties have a similar size to the modelling ones. ==> Yes, the sentence is part of the paragraph describing the inclusive cross section. The discussion of the differential cross section is in the following one.

EFT interpretation: - Is this performed using both channels? ==> Yes, clarified in the text: "the differential cross-section as a function of the photon pT from the combined measurement in the single-lepton and dilepton channels."

Fig 2: There is a mismatch between the top right plot and the caption "I(CtW)" vs "R(CtW)", please correct the caption. ==> Fixed!

---

## Editorial Decision

resubmitted